# Synthesis of Effect Sizes on Dose Response from Ultra-Processed Food Consumption against Various Noncommunicable Diseases

**DOI:** 10.3390/foods12244457

**Published:** 2023-12-12

**Authors:** Fairuz Firda Bestari, Nuri Andarwulan, Eny Palupi

**Affiliations:** 1Department of Food Science, Faculty of Agricultural Technology, IPB University, Bogor 16680, Indonesia; 2Department of Community Nutrition, Faculty of Human Ecology, IPB University, Bogor 16680, Indonesia

**Keywords:** dose response, meta-analysis, noncommunicable disease, ultra-processed foods, NOVA system

## Abstract

Ultra-processed foods (UPFs), according to the NOVA classification food system, are food products that are processed using advanced processing technology, which improves palatability and sensory quality. However, UPFs increase the daily intake of energy, sodium, sugar, and total fat (including saturated fat), and decrease the intake of fiber. This might trigger overweight and obesity, the initial stages of noncommunicable diseases (NCDs). However, the effect of UPF consumption on NCDs remains under debate. This study aimed to synthesize the effect size of UPF dose response on various NCDs by using a meta-analysis method. The main output was a hazard ratio (HR) and a 95% confidence interval (CI). Using the Preferred Reporting Items for Systematic Review and Meta-Analyses (PRISMA) selection guidelines, 18 research articles were chosen for further effect size synthesis. The results showed that UPF consumption significantly increased the daily intake of carbohydrates, added sugar, saturated fat, sodium, energy, cholesterol, and total fat; increases of 49.64, 40.78, 30.00, 27.76, 26.67, 25.69, and 15.77%, respectively, were observed. Moreover, with UPF consumption, the fiber intake was way below the daily dietary recommendation (DR), at −38.55%. Further, a 10% increment in UPF consumption significantly affects diabetes, cardiovascular diseases, obesity, and cancer (HR ± 95% CI: 1.115 ± 0.044, 1.096 ± 0.053, 1.068 ± 0.050, and 1.020 ± 0.020, respectively). Thus, limiting daily UPF consumption could help prevent obesity and various NCDs.

## 1. Introduction

Ultra-processed foods (UPFs), under the NOVA food classification system, are foods produced using advanced processing technology (involving heat) and/or containing food additives to enhance their sensory quality and shelf life [1]. However, 78.2% of consumers understand that UPFs are foods that have gone through many processes in the industry [2]. Currently, the variety and sales volume of UPFs are increasing in the world. The UPF sales volumes in Southeast and South Asia, and in North Africa and the Middle East have reached 67.3% and 57.6%, respectively [3]. This may be because UPFs have a longer shelf life, which helps prevent food loss and waste. In addition, UPFs often offer better sensory qualities (appearance, flavor, texture, and taste) than other NOVA food groups, thereby increasing palatability and acceptance [4]. With their good palatability, UPFs can be an alternative to overcome nutritional deficiencies, the world’s biggest problem. However, many UPFs contain high proportions of carbohydrates, fats, and sugar (51.2, 29.7, 21.1%, respectively), and salt (>600 mg/100 g), compared to other NOVA foods [5,6,7]. Therefore, excessive UPF consumption negatively impacts health.

Good palatability causes frequent overconsumption of food. Increased consumption of UPFs can trigger excessive calorie intake, causing the body fat percentage to increase. An increased body fat percentage can contribute to metabolic syndromes such as overweight and obesity, characterized by increased waist circumference (>101.6 cm and >88.9 cm, in men and women, respectively), blood pressure (>130/85 mmHg), fasting triglyceride (TG) levels (>150 mg/dL), fasting blood glucose levels (>100 mg/dL), and low fasting high-density lipoprotein cholesterol levels (<40 mg/dL and <50 mg/dL, in men and women, respectively) [8,9]. Metabolic syndrome is the initial stage of noncommunicable diseases (NCDs) such as diabetes, cancer, and cardiovascular disease [10,11,12]. In 2022, the World Health Organization, noted that NCDs can kill 41 million people annually, comprising around 74% of global mortality. Among NCDs, cardiovascular diseases have the highest mortality (17.9 million people per year), followed by chronic respiratory diseases (4.1 million people per year), and diabetes (2 million people per year, including deaths from kidney disorders due to diabetes). Therefore, analyzing the dietary patterns associated with the risk of excessive calorie intake is important.

Here, we aimed to reveal the relationship between NCD risk and the consumed amount of UPFs. Further, we aimed to estimate the appropriate allowable portion for UPF consumption. Here, using effect size synthesis methods we examined the nutritional contribution of UPFs to daily required allowance, and estimated the effect of a 10% increment in UPF consumption on various NCDs.

## 2. Materials and Methods

### 2.1. Literature Search and Selection

We followed the 2021 PRISMA (Preferred Reporting Items for Systematic Review and Meta-Analyses) guidelines in conducting the literature survey for effect size synthesis [13]. Briefly, a comprehensive literature search was conducted using various databases, including Google Scholar, ScienceDirect, Springer, ResearchGate, and PubMed. The keywords used in this study were optimized using Boolean operators [14]. The keywords included in this study were “processed food OR ultra-processed food”, “cancer OR carcinoma”, “diabetes”, “cardiovascular OR cardio”, and “obesity OR obese”.

### 2.2. Inclusion and Exclusion Criteria

Here, the following inclusion criteria for studies were used: (1) original article published within the last 20 years; (2) article focuses on the increased contribution of UPFs to NCDs; (3) article covers the parameters belonging to one or several of these NCDs (cardiovascular disease, obesity, diabetes, and cancer); (4) study participants are men and/or women aged ≥18 years; (5) article has cohort/case-control/cross-sectional study design; (6) article measures UPF consumption as a percentage of weight proportion, using food frequency questionnaires, food records, and 24-h dietary recall; and (7) article provides effect size data on hazard ratio, odds ratio, risk ratio. References were excluded if the data presented were incomplete. The research questions for this study were formulated using the Population, Intervention, Comparison, and Outcomes approach [15]. The formulated research questions were as follows: (1) “Is there any noticeable additional contribution of certain nutritional components when comparing high and low level of UPF consumption?”; (2) “Do increments in UPF consumption significantly affect the risk of NCDs?”.

### 2.3. Data Synthesis and Statistical Analysis

We meta-analyzed the effect size by applying the Hedges’ g (d_++_) and hazard ratio (HR) approach [16]. The contribution of UPFs to nutrient content (in terms of weight proportion) was reported as mean and standard deviation. Hedges’ g (d_++_) and 95% confidence intervals were applied as effect sizes for various parameters, including energy, sodium, total fat, saturated fat, carbohydrates, fiber, added sugar, and cholesterol. The effect size (d) was calculated as follows:(1)$$d=\frac{\bar{Xe}-\bar{Xc}}{S}J$$
where $\bar{X}$_e_ is the mean value from the experimental group (high consumption of UPFs) and $\bar{X}$_c_ is the mean value of the control group (low consumption of UPFs). S is pooled standard deviation, described as follows:(2)$$S=\sqrt{\frac{\left(Ne-1\right){\left(Se\right)}^{2}+\left(Nc-1\right){\left(Sc\right)}^{2}}{\left(Ne+Nc-2\right)}}$$

J is the factor correction for small sample size, and was calculated as:(3)$$J=\;1-\frac{3}{\left(4\left(Nc+Ne-2\right)-1\right)}$$
where Ne is the sample size for the experimental group, Nc is the sample size for the control group, Se standard deviation of the experimental group, and Sc is standard deviation of the control group. The variance of d is described as:(4)$$V_{d}=\frac{\left(Nc+Ne\right)}{NcNe}+\frac{d^{2}}{\left(2\left(Nc+Ne\right)\right)}$$

SE_d_ is standard error for (d), and was calculated as:(5)$${\mathrm{SE}}_{d}=\sqrt{V_{d}}$$

Bias in d can be eliminated by correction with Hedges’ g and calculated as:(6)g = J × d
(7)$$V_{g}=\;J^{2}\times V_{d}$$
and
(8)$${\mathrm{SE}}_{g}=\sqrt{V_{g}}$$

The cumulative effect size Hedges’ g (d_++_) is formulated as:(9)$$d_{++}=\frac{\sum_{i=1}^{n}w_{i}d_{i}}{\sum_{i=1}^{n}w_{i}}$$
where w_i_ is the inverse of the sampling variance.

The effect size of increments in UPF consumption and the prevalence of NCDs were expressed in odds ratio, risk ratio, and hazard ratio with the final conversion as hazard ratio and a 95% confidence interval (CI). Comprehensive meta-analysis software was used to convert data from odd and risk ratio into hazard ratio. The results from calculated effect sizes and confidence intervals (of each study) were transformed into a forest plot. Heterogeneity was calculated using the Cochran Q-Statistic and I^2^, which measures the level of difference between studies. Q was calculated as:(10)$$Q=\sum_{I}wi{\left(y_{i}-{\bar{y}}_{w}\right)}^{2}$$
where y_i_ is the estimated experimental group, $\bar{y}$_w_ = $\sum_{i}w_{i}y_{i}/\sum_{i}w_{i}$ is the estimated weight of the experimental group, and w_i_ is the inverse sampling variance-i. Then, I^2^ is formulated as:(11)$$I^{2}=\frac{Q-df}{Q}\times 100$$

Q is the value of x^2^ and df is the degrees of freedom (df = k − 1). Heterogeneity was considered significant if the Q statistic was significant (*p* < 0.01) or I^2^ > 50%. The effect size estimation was computed using a weighted random-effects model based on the DerSimonian–Laird approach. Publication bias was measured using Egger’s regression and Begg’s rank correlation [17]. Egger’s regression was used to examine the linearity between the estimated intervention effect and the standard error. Begg’s rank correlation was used to examine the relationship between the estimated intervention effect and the sampling variance. Results were considered statistically significant at *p* < 0.05. The fail-safe number represents the total number of studies needed to determine the significance of the average meta-analysis study [16].

### 2.4. Study Characteristics

The PRISMA flowchart of the selection stages of the research articles (papers) is presented in Figure 1. Eighteen papers that met the inclusion criteria were obtained after several article identification and screening steps and were used for effect-size synthesis (see Appendix A). Briefly, a preliminary literature survey found 74,233 papers on the topics “processed food OR ultra-processed food”, “cancer OR carcinoma”, “diabetes”, “cardiovascular OR cardio”, and “obesity OR obese” using various search engines (see Section 2.1). Then, firstly, duplicate research articles (numbering 32,614) were removed. Secondly, 40,807 (of the remaining 41,619) papers were removed because they did not contain the keyword “processed food”. Third, an additional 650 papers were excluded because their titles did not relate to NCDs. Fourthly, in the subsequent screening step, 162 papers were selected based on abstracts, but 51 research articles were excluded because the abstracts were not appropriate, leaving 111 articles. Of these, 93 articles were dropped because they did not meet all the criteria for selection; for example, either the full texts were inaccessible, the paper types were proceedings, editorials, or comments, or data on increments in UPF consumption were lacking. Thus, finally, at the end of the identification and selection steps, 18 articles (including 28 experiments/studies) covering several NCDs (diabetes, cancer, obesity, and cardiovascular disease) were finally chosen. Study sample sizes for diabetes, cancer, cardiovascular disease, and obesity were 5 [18,19,20,21,22], 12 [11,22,23,24,25], 5 [10,22,26,27,28], and 6 [12,20,29,30,31,32], respectively. 

## 3. Results and Discussion

### 3.1. Subject and UPF Product Profile

The subjects included in this study were 18–65 years of age, with an average BMI of 30.88 kg/m^2^. According to World Health Organization criteria, subjects with BMI 25–29.9 kg/m^2^ and BMI ≥ 30 kg/m^2^ were categorized as overweight and obese, respectively [33]. Thus, 51%, 29%, and 20% of subjects in this study had normal (with normal nutritional status), overweight, and obese BMIs, respectively. This indicated a high prevalence of overnutrition, particularly obesity, considering that the subject sampling was primarily intended to represent the region. This phenomenon indicated an aggravated risk of NCDs because a high BMI is correlated with a high risk of several NCDs, such as cardiovascular disease, high blood pressure, type 2 diabetes, and cancer [34].

Each subject in this study consumed a variety of UPFs (Appendix A), including items such as bread, pastries, cakes, biscuits, industrially processed food (chips, meat, fish, eggs), condiments and sauces, pizza, ultra-processed fruit and vegetables, margarine, added sugar, ultra-processed dairy products, salty snacks, and ready-to-eat/heat meals. Table 1 reveals that industrial bread, dessert, and beverages comprised 13.44, 13.00, and 12.98%, respectively, of their daily consumption. The percentage contribution of each food consumed is presented in Table 1.

The subjects in the reviewed studies had varying levels of UPF consumption (Appendix A), which were categorized into quartiles (Q1, Q2, Q3, Q4, Q5) from lowest into highest. Two consumption patterns emerged: Q1 (Lo) as the comparison group, with an average weight proportion of 3.4%, and Q5 (Hi) as the highest consumption pattern, with an average weight proportion of 93%. The total energy consumed amounted to 2533.36 kcal, exceeding the recommended daily intake of 2000 kcal.

### 3.2. Effect of UPF Consumption on Nutrient Intake 

High consumption of UPFs correlated with an increased intake of several nutrients—such as carbohydrates, total fat, saturated fat, sugar, salt, cholesterol, and fiber—closely associated with NCD risk. The forest plot of the effect size of UPF consumption on various nutrients is presented in Figure 2. The meta-analysis results indicated that high UPF consumption significantly promoted excess intake, especially of energy, followed by carbohydrates, total fat, added sugar, saturated fat, cholesterol, and sodium, with effect size (d_++_ ± 95% CI) 1.546 ± 0.180, 0.776 ± 0.24, 0.614 ± 0.101, 0.562 ± 0.204, 0.536 ± 0.718, 0.469 ± 0.174, and 0.341 ± 0.131, respectively. Conversely, high UPF consumption decreased the fiber intake by approximately 40%. Interpretation of the effect size was performed using Cohen’s benchmark to determine the impact of the effect size. Energy intake had a large effect, with an effect size greater than 0.8. This was followed by the effect size of carbohydrates and total fat with moderate effect size. Effect size is considered small if the value is <0.2, moderate at 0.5, and large if >0.8 [35,36].

#### Gap in the Contribution of UPFs Consumption on Dietary Recommendation in Weight Proportion

The results of the synthesized effect size ± 95% CI, nutrient contribution, and % gap compared to the daily recommended standard are shown in Table 2. When compared to the daily recommended standards, both the high and low UPF consumption groups had excessive energy, sodium, carbohydrates, total fat, saturated fat, added sugar, and cholesterol intakes. The lowest UPF consumption group (Lo) showed energy, sodium, carbohydrates, total fat, saturated fat, fiber, added sugar, and cholesterol intakes of 1672.16 kcal/d, 2245.28 mg/d, 325.98 g/d, 63.18 g/d, 21.57 g/d, 20.13 g/d, 51.43 g/d, and 205.95 mg/d, respectively. In contrast, the highest UPF consumption group (Hi) showed energy, sodium, carbohydrates, total fat, saturated fat, fiber, added sugar, and cholesterol intakes of 2533.36 kcal/d, 2555.13 mg/d, 441.44 g/d, 74.20 g/d, 26.13 g/d, 19.05 g/d, 70.39 g/d, 251.38 mg/d, respectively. 

Effect size synthesis results indicated that the Hi group significantly exceeded the standard recommended intake of carbohydrates, added sugar, saturated fat, sodium, cholesterol, energy, and total fat by 49.64, 40.78, 30.00, 27.76, 25.69, and 15.77%, respectively. Moreover, both groups had low fiber intakes. UPF consumption caused a very low fiber intake (19.05 g/day), which is 38.55% below the recommended level. The synthesized effect size results strongly indicate that UPF consumption robustly affects the studied parameters, which is supported by the fail-safe number (N_fs_) (symbolized by R in the table) [36]. 

Consumption of UPFs significantly increases sugar intake. This is because UPFs contain eight times more added sugar content than other processed foods. UPFs and processed foods contribute (up to) 21.1 and 2.4% of the daily recommended sugar intake, respectively [6]. Excessive sugar intake increases the workload of the pancreatic enzymes. Over extended periods of time, this might reduce their sensitivity to blood glucose and lead to the development of diabetes. Respondents with a regularly intake of high amounts of carbohydrates and sugar showed a significant increase in their blood glucose levels [37].

Table 2 shows that UPF consumption increased saturated fat intake by 30% over the recommended dietary intake. This might be attributed to the high levels of total and saturated fat found in UPFs. UPFs contain higher amounts of total and saturated fat than minimally processed foods (MPFs), such as fruit and vegetables. For example, UPFs and MPFs contain (per 100 g) 4.0 and 1.5 g fat, respectively [38].

UPFs are suspected to contain high amounts (>1.5 g/100 g) of sodium [39]. The sodium content of UPFs is particularly high in various products; chili sauce, soy sauce, and instant noodles contain (per 100 g) 1854, 1319, and 1257.31 mg sodium, respectively. Additionally, local products contain higher sodium (78.86%) than imported products (21.14%) [40].

### 3.3. Hazard Ratio Associated with Consuming UPFs and Developing NCDs

UPFs, depending on the product, have different size effects on the risk of NCDs. However, a 10% increase in the consumption of UPFs per day is associated with an increased risk of obesity (HR = 1.068; 95% CI = 1.018–1.121; I^2^ = 87.574; *p*-value = 0.007), diabetes (HR = 1.115; 95% CI = 1.071–1.161; I^2^ = 22.971; *p*-value = 0.000), cancer (HR = 1.021, 95% CI = 1.002–1.040; I^2^ = 67.745; *p*-value = 0.032), and cardiovascular disease (HR = 1.096; 95% CI = 1.043–1.151; I^2^ = 74.482; *p*-value = 0.001). Increased risk is associated with increased UPF consumption for premenopausal breast cancer (HR = 1.015; 95% CI = 0.952–1.083) and postmenopausal breast cancer (HR = 1.052; 95% CI = 0.935–1.184). Notably, obesity parameters exhibit a higher heterogeneity in NCD risk association than the other parameters studied here. Thus, significant differences in results may be expected among studies analyzing obesity and NCD risk. In contrast, the diabetes group showed the lowest heterogeneity in association with NCD risk. A forest plot is presented in Figure 3.

Obesity occurs when daily energy intake exceeds recommended levels. Between 60 and 80% of surplus energy is stored as fat, primarily in the form of glycogen. When energy intake consistently surpasses this limit, fat accumulation in the body increases as excess energy is continually stored in adipose tissue [41].

The effect size showed that a 10% increase in daily UPF consumption was associated with the greatest risk of diabetes. This association may be attributed to an imbalance in nutrition due to frequent consumption of sugary foods. Further, this increased risk of diabetes can also be attributed to various factors, including the nutritional content of UPFs, characterized by high sugar and energy intakes, which often exceed the recommended levels [42,43]. Furthermore, people’s preferences in reading nutritional facts and other important details about food products affect UPF intake. Mostly, they prioritize taste and price over noting ingredients (including caloric values and other listed ingredients) [44]. 

Excessive sodium in the bloodstream will bind a large amount of water. This leads to an increase in blood volume and the occurrence of hypertension, eventually contributing to coronary heart disease [45]. Each one-gram increase in sodium intake can increase the risk of cardiovascular disease by 6% [46]. Establishing a daily limit for sodium intake by reading product ingredient labels carefully and by stipulating industry standards for permissible salt content can help mitigate this risk. Further, excessive sodium intake may lead to cardiovascular disease by increasing triglyceride and low-density lipoprotein cholesterol levels, which are risk factors for cardiovascular diseases [47].

Poor dietary patterns are the most common cause of cancer. Excessive fat consumption can exacerbate this risk by contributing to the formation of peroxides and other reactive oxygen species which can cause DNA damage. Additionally, a high fat diet can increase release of sterol metabolites into the intestines, which can cause tumorigenesis in the colonic epithelium [48].

#### 3.3.1. Cumulative Effect Size of Ultra-processed Foods and Non-Communicable Diseases

Table 3 shows the hazard ratio as an effect size, a 95% confidence interval, and a fail-safe number. NCD parameters have robust (R) fail-safe number, indicating that it is safe to derive conclusions based on them. Each NCD has a *p*-value < 0.001, except for obesity, which has a *p*-value < 0.01.

#### 3.3.2. Publication Bias of Meta-Analysis on UPFs and NCDs

Publication bias was interpreted using Egger’s regression and Begg’s rank correlation. A *p*-value ≥ 0.05 means absence of bias [17]. For diabetes, the *p*-values for Egger’s regression and Begg’s rank correlation tests were 0.141 and 0.290, respectively. For cancer, the *p*-values of Egger’s regression and Begg’s rank correlation tests were 0.205 and 0.411, respectively. Thus, (all *p*-values ≥ 0.05 indicate that) there was no publication bias for cancer and diabetes. However, for cardiovascular disease, the *p*-values of the Egger’s regression and Begg’s rank correlation tests were 0.018 and 0.086, respectively. Further, for obesity, the *p*-values of the Egger’s regression and Begg’s rank correlation tests were 0.012 and 0.347, respectively. Thus, there was a small publication bias when analyzing associations of UPF consumption with risks of developing obesity and cardiovascular disease. 

## 4. Conclusions

Our results from the meta-analysis of publications in the last two decades show that UPF consumption significantly increases the intake of carbohydrates, added sugars, saturated fat, sodium, and cholesterol. Notably, the group that consumed high levels of UPFs exhibited 39% lower fiber intake than the low UPF consumption group. This study also demonstrates that 10% increment in the consumption of UPFs significantly increases the risk of several NCDs, such as diabetes, cardiovascular disease, obesity, and cancer. This highlights the importance of paying attention to UPF consumption and maintaining a balanced nutritional diet. Furthermore, there may be role for government regulations thatwhich limit sugar, sodium, fat, and food additives during UPFs production in preventing high NCD rates. This study is limited by a low sample size of studies in certain subgroups; this has not been addressed here. Future studies similar to this one could support the conclusions of this study, especially concerning further evaluation of subgroup analysis. Moreover, this study can be extended to include the influences of various other risk factors for NCDs, such as physical activity, smoking status, sleep patterns, stress management, alcohol consumption, nutritional status, etc., in order to comprehensively understand NCDs. The risk factors of NCDs could be included as a moderator variable in the meta-analysis. 

## Figures and Tables

**Figure 1 foods-12-04457-f001:**
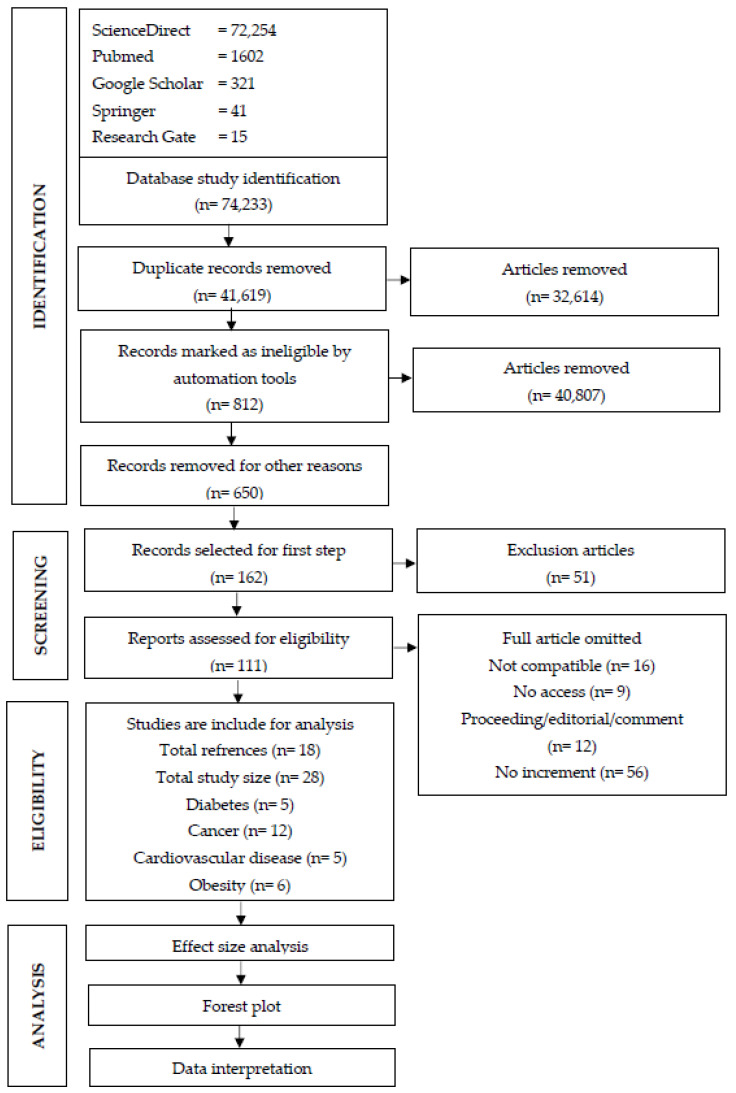
Steps for screening article with PRISMA flow diagrams for meta-analysis of UPF consumption and NCDs.

**Figure 2 foods-12-04457-f002:**
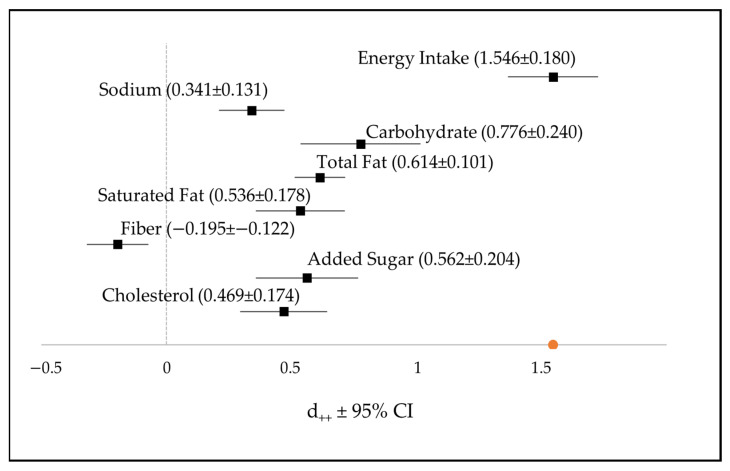
Forest plot of the contribution of UPFs to the intake of energy, sodium, carbohydrates, total fat, saturated fat, edible fiber, added sugar, and cholesterol in weight proportion. The values indicate Hedges’ g cumulative effect size (d_++_) and a 95% Confidence Interval (CI).

**Figure 3 foods-12-04457-f003:**
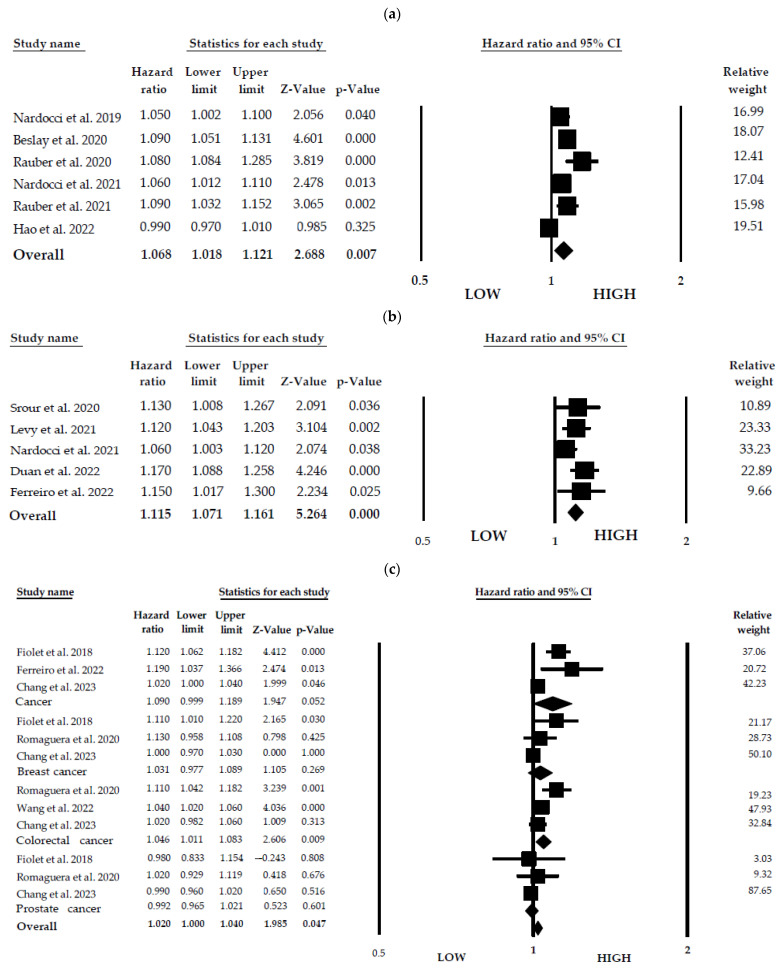
Forest plot of dose–response 10% increment UPFs and (**a**) obesity; (**b**) diabetes; (**c**) cancer; (**d**) cardiovascular diseases [10,11,12,18,19,20,21,22,23,24,25,26,27,28,29,30,31,32].

**Table 1 foods-12-04457-t001:** Food type contribution (% of total energy).

FOOD TYPE (% of Total Energy)	AUTHORS									AVERAGE (%)
	[11]	[12]	[26]	[19]	[30]	[22]	[31]	[28]	[24]	
Cereal	16	2.67	12	0.6	4.2	4.5	6	34.5		10.06
Ultra-processed fruits and vegetables	15	5.07	18					4.6		10.67
Margarine	26	3.94		0.6	2.1	1		4.6		6.37
Sugary products		6.82	28	0.4	3.2			6.4		8.96
Ultra-processed dairy products	7	0.43	8			16.5		8.4	4.5	7.47
Salty snacks	2	2.36	2	0.3	1.6	1		8.4	4	2.71
Meat, fish, poultry	6	2	11	1.2	3.6	16.3		8.6	4	6.59
Sauces and dressings	2	4.61	5	0.3	2.2	2	9	8.7	22	6.20
Beverages	20	6.73	16	5.8	2.1	18.4	15	15.8	17	12.98
Industrial bread		8.31			11		21			13.44
Frozen/ready-to-eat/heat meals				1.1	7.8		16		4.5	7.35
Desserts				2.8	3.2	20.5	21			13.00
Biscuit				0.6	3.2					1.90
Industrial chips				3.4	2.5				26	10.63
Industrial pizza				0.6	1.5					1.05
TOTAL	94	42.94	100	17.70	48.20	80.20	88	100	82	

**Table 2 foods-12-04457-t002:** Cumulative effect size, weighted mean value, fail-safe number, % additional contribution, and *p*-value of nutrient parameters.

Parameters	Unit	d_++_	±95% CI	Studies Size (N)	N_fs_	Hi-Mean	Lo-Mean	DR	% AC	SEM	*p*-Value
Energy intake	kcal/day	1.546	0.180	35	37,430 ^R^	2533.36 ^b^	1672.16 ^a^	2000	26.67	0.092	***
Sodium	mg/day	0.341	0.131	19	42,427 ^R^	2555.13 ^b^	2245.28 ^a^	2000	27.76	0.067	***
Carbohydrate	g/day	0.776	0.240	16	93,009 ^R^	441.44 ^b^	325.98 ^a^	230–360	49.64	0.122	***
Total fat	g/day	0.614	0.101	20	10,751 ^R^	74.20 ^b^	63.18 ^a^	50–75	15.77	0.051	***
Saturated fat	g/day	0.536	0.178	10	1032 ^R^	26.13 ^b^	21.57 ^a^	20	30.00	0.090	***
Fiber	g/day	−0.195	−0.122	22	3719 ^R^	19.05 ^a^	20.13 ^b^	25–37	−38.55	0.063	***
Added Sugar	g/day	0.562	0.204	14	12,732 ^R^	70.39 ^b^	51.43 ^a^	50	40.78	0.104	***
Cholesterol	mg/day	0.469	0.174	7	2040 ^R^	251.38 ^b^	205.95 ^a^	200	25.69	0.089	***

Notes: d_++_, cumulative effect size; CI, Confidence Interval; N, study size for effect size calculation; N_fs_, fail-safe number; Hi-mean, mean value from high consumption group; Lo-mean, mean value from low consumption group; DR, Dietary Recommendation; % AC, % ∆ Contribution gathered from the gap between Hi-mean UPFs consumption with the median of DR at % unit; SEM, standard error mean; R, robust (N_fs_ > 5N + 10); a, low value; b, high value; ***, *p*-value < 0.001.

**Table 3 foods-12-04457-t003:** Cumulative effect size, fail-safe number-weighted mean value, and *p*-value of NCD parameters.

Noncommunicable Diseases	Hazard Ratio	±95% CI	Study Size (N)	N_fs_	*p*-Value
Obesity	1.068	0.050	6	53 ^R^	**
Diabetes	1.115	0.044	5	45 ^R^	***
Cancer	1.020	0.020	12	86 ^R^	*
Cardiovascular Disease	1.096	0.053	5	61 ^R^	***

Notes: CI, Confidence Interval; N, study size for effect size calculation; N_fs_, fail-safe number; R, robust (N_fs_ > 5N + 10); *, *p*-value < 0.05; **, *p*-value < 0.01; ***, *p*-value < 0.001.

## Data Availability

Data is contained within the article.

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
