# Peer review of "Synthesis of Effect Sizes on Dose Response from Ultra-Processed Food Consumption against Various Noncommunicable Diseases"

_foods, 2023, doi:10.3390/foods12244457_

Round 1

Reviewer 1 Report

Comments and Suggestions for Authors

1. The purpose and significance of this study need to be further deepened in the introduction.

2. There are a lot of formulas in this paper, and the serial number of formulas needs to be marked.

3. The research method lacks reference support.

4. It is difficult to draw convincing conclusions just by summarizing and analyzing the existing data. Foods Journal as a high level of research journal, the study did not reach the level of Foods journal.

Author Response

Dear the Referees of Foods,

Thank you very much for the feedback of our manuscript and for giving us a chance to carry out revision along the comments given by the reviewers. Below please see the attachment. All line numbers given refer to the revised version of the manuscript. All the changes are marked with blue highlight for revision in the manuscript.

With kind regards, on behalf of all authors,

Eny Palupi

Reviewer 2 Report

Comments and Suggestions for Authors

1. line 41 introduction inches must be transformed in cm, in scientific publications we all use SI units

2. lines 345-360, English MUST be improved, there are missing words, missing spirit of the language, looks like google translator

3. lines from 385- 411 English must be improved. 

4. line 407 dietary intake is the most common cause of cancer, the question is dietary intake of what? if authors mean in general what we eat, than different term should be used, like nutrition or nutrition habits. On the other hand as a statement this is not true, because cancer pathogenesis is so complex and unexplored that I think this sentence should be rephrased.

5. line 556 reference is written in pictograms (chinese or japanese) I cant tell, and it must be somehow translated in english, otherwise is uslless.  Author are written in latinic, maybe authors should contact them for transcription. 

Comments on the Quality of English Language

2. lines 345-360, English MUST be improved, there are missing words, missing spirit of the language, looks like google translator

3. lines from 385- 411 English must be improved. 

Author Response

(The authors gave the same response as above.)

Reviewer 3 Report

Comments and Suggestions for Authors

Food processing in any general sense is not a public health issue. The full impact of industrialized food processing on dietary patterns, including the environments of eating and drinking, remains overlooked and underestimated. Many forms of food processing are beneficial. Ultra-processed foods are products made mostly or entirely from substances extracted from foods or derived from food constituents with little if any intact food, which often contain flavours, colours and other additives that mimic or intensify the sensory qualities of foods or culinary preparations made from foods.

The basic purpose of dose-response meta-analysis (DRMA) is to reveal the relationship between disease risk and exposure dose. A forest plot is a useful graphical display of findings from a meta-analysis. It provides essential information to inform our interpretation of the results. Typically, a forest plot contains six basic columns (included studies, intervention group, control group, weight, outcome effect measure in numeric format, outcome effect measure in graphical presentation).

Non-communicable diseases are diseases that are not spread through infection or through other people, but are typically caused by unhealthy behaviours. They are the leading cause of death worldwide and present a huge threat to health and development. Four types of non-communicable diseases account for over two thirds of deaths globally: cardiovascular diseases, cancers, diabetes and chronic respiratory diseases.

This study aimed to synthesize the effect sizes on dose-response of ultra-processed food consumption on various non-communicable diseases by using a meta-analysis method.

Recommendations

L 436 Please present the strengths and the limitations of your study.

Author Response

(The authors gave the same response as above.)

Reviewer 4 Report

Comments and Suggestions for Authors

Dear Authors,

What I miss in your study is a reference to tools like The Newcastle-Ottawa Quality Assessment Scale, often referred that are used for assessing the quality of non-randomized studies, such as cohort studies and case-control studies, in systematic reviews and meta-analyses.

How the UPF was defined in the studies you included in the analysis? Was there a reference made to NOVA classification in all studies?

How consumption of UPF was measured? Is there any potential source of bias?

Records removed for other reasons (n=650) - provide explanation

How the study design you included: cohort studies, case-control studies, and cross-sectional studies could influence your results? What about information bias, as a consequence of self-reported data on dietary intake?

Reflect more on limitation of your study

Author Response

(The authors gave the same response as above.)

Round 2

Reviewer 1 Report

Comments and Suggestions for Authors

None.